# Towards mouse genetic-specific RNA-sequencing read mapping

**Nastassia Gobet** [1,2], **Maxime Jan** [1,3], **Paul Franken**[1], **Ioannis Xenarios** [4,5]*

**1** Centre for Integrative Genomics, University of Lausanne, Lausanne, Switzerland, **2** Vital-IT, Swiss Institute of Bioinformatics, Lausanne, Switzerland, **3** Bioinformatics Competence Center, University of Lausanne, Lausanne, Switzerland, **4** Ludwig Cancer Research/CHUV-UNIL, Lausanne, Switzerland, **5** Health 2030 Genome Center, Geneva, Switzerland

\* ioannis.xenarios@unil.ch

## Abstract

Genetic variations affect behavior and cause disease but understanding how these variants drive complex traits is still an open question. A common approach is to link the genetic variants to intermediate molecular phenotypes such as the transcriptome using RNA-sequencing (RNA-seq). Paradoxically, these variants between the samples are usually ignored at the beginning of RNA-seq analyses of many model organisms. This can skew the transcriptome estimates that are used later for downstream analyses, such as expression quantitative trait locus (eQTL) detection. Here, we assessed the impact of reference-based analysis on the transcriptome and eQTLs in a widely-used mouse genetic population: the BXD panel of recombinant inbred lines. We highlight existing reference bias in the transcriptome data analysis and propose practical solutions which combine available genetic variants, genotypes, and genome reference sequence. The use of custom BXD line references improved downstream analysis compared to classical genome reference. These insights would likely benefit genetic studies with a transcriptomic component and demonstrate that genome references need to be reassessed and improved.

**Data Availability Statement:** All relevant data are within the manuscript and its Supporting Information files. Raw reads files are accessible on NCBI Gene Expression Omnibus under accession id GSE114845. Given that data sources are diverse

## Author summary

To understand how genetic variations affect behavior and cause disease it is common to quantify expression of transcripts by sequencing. Transcripts are extracted, fragmented, and the sequence of the fragments read. An important step for their quantification is to virtually assign the different fragments to the transcript they originate from using a reference genome. Reference genomes are costly to build, so usually only one high-quality reference per animal model species is available. When comparing genetically different individuals, using a single reference may introduce a bias because it might be more similar to some individuals than to others. Paradoxically, the variations at the core of genetic studies are thus ignored at the start of the analysis. We built customized references with known genetic variants for each of the mouse lines we had and quantified the impact of the reference at different levels of the bioinformatic analysis. We found that using customized references reduced the bias compared to using a single reference. Our study uses

and some do not have a version or identifier, we grouped data so that it is easier to find (10.5281/zenodo.5513980). This repository contains input data: D2-specific variants from dbSNP (vcf), genotypes from GeneNetwork (tab); intermediate and produced data: D2 blocks (bed), imputed variants (vcf), BXD-specific genome sequences (fasta), BXD-specific transcriptome annotation (gtf), gene counts, normalized gene expression, local eQTLs. The code used for the analyses is on https://github.com/nagobet/BXDmapping.

**Funding:** P.F. and M.J. were funded by the University of Lausanne (Etat de Vaud). N.G. was funded by the Swiss National Science Foundation grant to P.F. (31003A_173182 and 310030B_192805, https://www.snf.ch/en). The funders had no role in study design, data collection and analysis, decision to publish, or preparation of the manuscript.

**Competing interests:** The authors have declared that no competing interests exist.

publicly available data and tools, so others can easily implement this improvement in their analyses.

## Introduction

To decipher how genome leads to phenome, measuring gene expression by RNA-sequencing (RNA-seq) is widely used. Fragments of RNA are read and then virtually mapped back onto a reference genome to determine the transcriptomic location of origin. Read mapping is often regarded as trivial but relies on many choices. Indeed, the user decides for example which reference to use and how exact the alignments are required to be. Most of the time, little information is published on how these choices are made. The mapping needs to account for amplification and sequencing errors, and for repeated sequences within the genome. It is important that the reference precisely represents the samples to guide the mapping. However, generating a reference assembly is complex and expensive and it is common practice to map all samples of a model organism to a single assembly provided by the genome reference consortium (GRC) [1,2]. The expression of non-reference alleles may be altered compared to that of reference alleles. This reference bias on the transcriptome can then spread to downstream analyses such as expression quantitative trait loci (eQTL) detection, where gene expression is associated to genomic variants. The genomic variations between individuals at the core of genetic studies are thus paradoxically often ignored at the start of the analysis and may alter interpretations and conclusions.

The genetic characteristics of humans have been widely studied and reference bias is known to alter DNA-seq, RNA-seq [3], and chromatin immunoprecipitation (ChIP)-seq analyses [4,5]. The ideal solution would be to use a sample-specific genome assembly. Since this is currently too costly, many methods to reduce reference bias were proposed [3,6,7]. One strategy notably used by the Genotype-Tissue Expression (GTEx) consortium is to tailor the analysis to each individual as initially implemented in the *WASP* suite of tools [8]. The WASP-correction proposes to map reads to the GRC assembly and identify mapped reads that overlap SNVs, then re-map these reads after replacing the reference alleles by variant alleles in the assembly and discard reads that change mapping loci. Although this strategy removes reference bias it also discards reads that are potentially informative of a genetic effect. Nevertheless, the idea of modifying the GRC reference assembly with variants specific to the individual or sample is used by many tools, with the difference that all the reads are mapped to the customized references only. For example, the AlleleSeq pipeline was developed for human trios where the variants for the two parents are known [9]. One of its tools, *vcf2diploid*, constructs two haplotype-specific references from one reference assembly and a list of genomic variants which can include single nucleotides variants (SNVs), indels, and structural variants (SVs). The authors proposed to map the offspring sample separately onto its two parental references, and to retain for each read the alignment with the highest alignment score. In case of equality the alignment is randomly taken from that of either parent to avoid systematic bias. *RefEditor* offers a similar approach, but adds a genotype imputation option [10]. Many tools aim at making the best use of large-scale variants and genotypes databases by genotype imputation to have for each individual a more complete set of alleles. However, these genotype imputation strategies cannot be applied to mouse or other animal models because of a lack of genetic characterization at the individual level.

Mouse genetic research mostly uses inbred lines, in which individuals are presumed isogenic. Therefore, it seems logical to aim for reference customization for mouse strains rather

than for individuals. The GRC mouse assembly is mainly based on the inbred strain C57BL/6J (B6) [11] and short genomic variants for many other inbred strains are available at dbSNP [12]. To compare retinal transcriptomes in two inbred strains (DBA/2J (D2) and B6), Wang et al. modified the GRCm38 reference genome with D2-specific variants from dbSNP to map the D2 samples [13]. This improved slightly the mappability by reducing the fraction of unmapped reads. *Seqnature* software aims at producing individualized diploid references for RNA-seq analysis and was used on simulated and real world data of Diversity Outbred (DO) mice, in which each mouse is a unique combination of 8 founder strains [14]. It shows improvement of the number of reads mapped, of the accuracy of transcript expression estimates, and of the number of eQTLs detected. Unfortunately, this type of study is very rare and the R package (DOQTL) used for the QTL analyses is specific to this mouse population, which renders comparisons with studies on different populations challenging. The Mouse Genome Project tries a more global approach to characterize the genetic variation among mouse strains. Many genetic variants were discovered and strain-specific genome assemblies for sixteen mouse strains were released [15]. However, it remains unclear how to use these resources for mice that are intercrossed.

The BXD panel of recombinant inbred lines is a well-studied and genetically simple population derived from the B6 and D2 strains [16]. Each BXD line has genetic markers (genotypes) available on the GeneNetwork website (http://genenetwork.org). Although this panel has been used in hundreds of studies, nobody to our knowledge has performed neither BXD-specific read mapping, nor BXD genome assembly. Here, we explored different strategies using publicly available resources to accurately represent the genetic diversity of the samples. We assessed the influence of the reference used for read mapping in this panel and how it impacts read mappability, gene expression, and eQTLs. We also measured how various parameter settings would influence the number of eQTLs found. We evaluated the use of the two parental genome assemblies and found this strategy not adequate. We implemented an alternative strategy which enhanced the GRC assembly with known variants. This improved the quality of BXD transcriptomics analyses. Our approach reduces reference bias in the BXD transcriptomics, and raises awareness about pitfalls of RNA-seq analyses.

## Methods

### Ethics statement

The authorization was given by the veterinary authorities of the state of Vaud, Switzerland (SCAV authorization #2534) as previously described in [17]. No sequencing data were collected specifically for this study.

### Samples and RNA-sequencing

We used RNA-seq samples obtained from the liver and the cerebral cortex of male mice from 33 BXD/RwwJ lines, the two parental strains C57BL/6J (B6), DBA/2J (D2), and their reciprocal F1 offspring (Fig 1A). The two tissues were collected at zeitgeber time (ZT) 6 (i.e., 6h after light onset) in mice that were either sleep deprived (SD) for the preceding 6 hours (ZT0-6) or mice that could sleep ad libitum (i.e. non-sleep deprived or NSD) [17,18]. Prior to sequencing, the RNA was pooled by mouse line and experimental condition (NSD or SD), such that material for a maximum of 3 mice contributed to each sample. Single-end reads of 100 bp were obtained using Illumina HiSeq 2500 system. A list of samples, including which BXD lines were used, is available (S1 Table). All samples passed quality control [17]. The eventuality of a mix-up of samples between strains, was tested previously by comparing the similarity between RNA-seq variant calling and GeneNetwork genotypes (see Fig 4 in [18]).

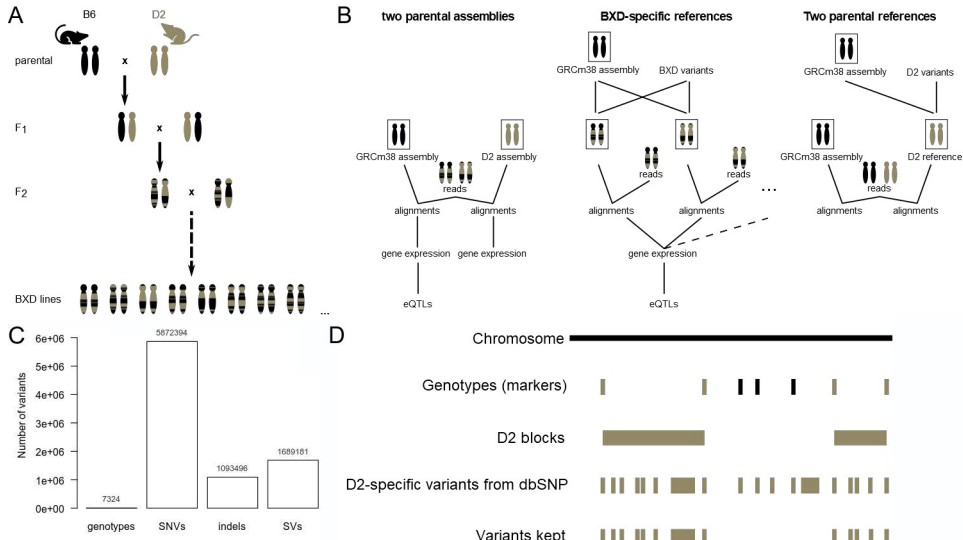

**Fig 1. Overview of strategies to utilize genomic variants in transcriptome read mapping in inbred mouse lines.** A. BXD mouse recombinant inbred panel. Samples came from mice that are: BXD advanced recombinant inbred lines, their parental inbred strains; i.e., C57BL/6J (B6) and DBA/2J (D2), and first generation cross between the parental strains (F1). B. The 3 RNA-seq read mapping strategies used in this study. In the 'two parental assemblies' strategy (left), the reads of all samples were mapped to the classical mouse genome assembly (GRCm38 or mm10) and to the D2 assembly. The 'BXD-specific references' (middle) were made from GRCm38 and BXD-specific variants. There is one reference for each BXD line, and the reads of each sample were mapped to the corresponding reference. The 'two parental references' (right panel) is an intermediate strategy in which the D2-specific reference was built from GRCm38 assembly and D2-specific variants. C. BXD genotypes available from GeneNetwork (genotypes) and D2-specific genomic variants (SNVs, indels, SVs) available from dbSNP. D. Genotypes imputation workflow. D2 haplotype blocks were delineated based on available genotypes in the BXD lines. D2-specific variants within these D2 blocks were included in the BXD-specific references. B6 regions or alleles are in black, D2 regions or alleles are in brown.

## Genome assemblies and transcriptome annotation download

Two genome assemblies and two transcriptome annotations were downloaded from Ensembl release 94 (ftp://ftp.ensembl.org/pub/release-94). The classical genome sequence GRCm38, also referred to as mm10, is based on the B6 strain (Mus_musculus.GRCm38.dna_sm.primary_assembly.fa). As D2 assembly the DBA/2J v1 genome sequence was used (Mus_musculus_dba2j.DBA_2J_v1.dna_sm.toplevel.fa) [15]. Both genome sequences do not contain alternative haplotypes, and repeats or low complexity regions are soft-masked (sm), which means they are represented as lowercase letters. Summary statistics of the assemblies were calculated in GAAS toolkit (https://github.com/NBISweden/GAAS, S2 Table). The transcriptome annotations correspond to these two assemblies (Mus_musculus.GRCm38.94.gtf and Mus_-musculus_dba2j.DBA_2J_v1.94.gtf).

## Variants download and genotype imputation

The BXD genotypes were downloaded from GeneNetwork. These are the alleles for each BXD line for 7324 genetic markers, which are variants selected to be indicative of recombination events between the parental genomes (BXD_Geno-19Jan2017_forGN.xlsx, Fig 1C). The D2-specific variants, which are 5'872'394 SNVs (DBA_2J.mgp.v5.snps.dbSNP142.vcf.gz) and 1'093'496 indels (DBA_2J.mgp.v5.indels.dbSNP142.normed.vcf.gz) from dbSNP (version 142, variants version 5), were downloaded (Fig 1C). To have a more complete set of genetic variants specific to each of the BXD lines, we performed genotype imputation as follows (Figs 1D and S3):

1. With the GeneNetwork BXD genotypes we defined for each BXD line the D2 haplotype blocks, as sets of at least 2 consecutive genotypes with D2 alleles without B6 or heterozygous alleles in between.

2. We checked which dbSNP D2-specific variants overlapped with the D2 blocks using bedtools (version v2.28.0).

3. We imputed the D2-specific variants overlapping with D2 blocks of a BXD line to be D2 alleles for this specific BXD line.

During this study, we noticed that genotypes from GeneNetwork for BXD100 (based on GRCm38 genome assembly also called mm10) had multiple chromosomes without any D2 alleles, which was unexpected considering this was not the case in the previous version of genotypes (based on MGSCv37 genome assembly also called mm9). GeneNetwork has been informed and the error was thought to have occurred during lift-over of the genotypes. We did not try to correct this mistake and kept the erroneous BXD100 genotypes in the current analysis. The effect, if any, on the eQTL analysis should be small, as this concerned only one BXD line out of 33, and not all chromosomes were affected.

## Customization of references

We built a customized reference genome for each BXD based on the GRCm38 assembly and BXD-specific genotypes (from GeneNetwork and imputed). For this, the reference genome sequence GRCm38 was customized for each BXD line with BXD-specific genotypes (from GeneNetwork and imputed) using vcf2diploid software (version 0.2.6) with a slight modification to change the software's behaviour with unphased heterozygous variants. Prior to compiling the software according to the installation instructions, we removed the function that randomizes unphased heterozygous variants (to determine whether there are included in the paternal or maternal genome) and the call to this function (see Table 1). It is, however, entirely possible to use the software without these modifications.

In the modified software, all unphased heterozygous variants were included in the genome sequence called "maternal" but ignored in the one called "paternal". We used the paternal sequence so that heterozygous genotypes were ignored. Note that all D2-specific variants from dbSNP and the BXD markers from GeneNetwork are unphased and heterozygous labels may be indicative of low or uncertain quality. On GeneNetwork (January 2017), genotypes are defined as "H (heterozygous) if the genotype was uncertain". In the 33 BXD lines we used, 1449 loci had H alleles out of the 7320 genotypes (20%) and on average 60 H alleles out of 7320 loci or a total 1967 H alleles out of 241560 alleles (0.82%). For D2-specific variants from dbSNP (version 142), Het means "Genotype call is heterozygous (low quality)". Of the

**Table 1. Modifications to vcf2diploid software.**

| File name | Code removed |
|---|---|
| Variant.java | public void randomizeHaplotype()<br>{<br> if (_rand.nextDouble() > 0.5) return;<br> int tmp = _paternal;<br> _paternal = _maternal;<br> _maternal = tmp;<br> return;<br>} |
| VCF2diploid.java | if (\!var.isPhased()) var.randomizeHaplotype(); |

5'872'394 SNPs 481'158 SNPs were "Het" (8%) and of the 1'093'496 indels 80'075 were Het (7%).

We also built a D2-specific reference genome based on the GRCm38 assembly and D2-specific SNPs and indels from dbSNP (Fig 1B "2 parental references"). We refer to this modified version as D2 reference, which differs from the D2 assembly in that the D2 assembly was assembled from DNA reads obtained in the D2 strain, whereas the D2 reference was a modified version of the assembly based on the B6 strain. We adapted the coordinates of the transcriptome annotation to the new coordinates for BXD and D2 references using the chain files generated by vcf2diploid and the liftOver tool (version 8.28) from UCSC (http://genome.ucsc.edu).

## Read mapping and setting of mapping parameters

We performed read mapping with STAR (version 2.7.0e) [19] with different values for the parameters. The default values were used for permissive alignment, whereas more restrictive alignments were obtained by varying the settings of the parameters: "*--scoreDelOpen -40*" to prevent deletions, "*--scoreInsOpen -40*" to prevent insertions, "*--alignIntronMax 1*" to prevent introns (splicing), "*--alignEndsType EndToEnd*" to prevent partial alignment of the read, and "*--outFilterMismatchNmax 0*" to prevent mismatches (the value is the maximal number of mismatches allowed). We also varied the inclusion (with annotation) or exclusion (without annotation) of the transcriptome annotation in the genome index.

To count the uniquely mapped reads per gene after STAR, HTseq (version 0.6.1p1) was used with samtools (version 1.9) to convert alignments from bam to sam format. Only the alignments with a quality score of 10 or above were kept (default). The command used was:

samtools view -h alignment.bam | htseq-count -s reverse -t exon -m union-reference.gtf

The "-s reverse" parameter was used for the stranded library which is specific to the library preparation and sequencing protocol. Alternatively, for mapping using transcriptome annotation, the HTseq counting implemented in STAR was used with --quantMode GeneCounts.

## Filtering and normalization of gene counts

Lowly expressed genes were filtered by tissue keeping only genes with counts per million (cpm) above 0.5 (min_cpm) for at least 20 samples (on a total of 66 BXD samples/tissue). Counts were normalized with the edgeR package (version 3.24.3) which uses the weighted trimmed mean of M-values (TMM) method to take into account the variation in library size and in RNA population [20] and log transformed (log2).

## Differential mapping analysis of genes

To compare the impact of the reference on the gene expression, duplicated gene names were removed, and only gene names common to both GRCm38 and D2 or GRCm38 and BXD references transcriptome annotation were kept. A differential expression analysis was performed on the BXD samples using the voom function from R package limma (version 3.38.3). Notice that in each case, the two groups compared had exactly the same samples, so only the reference used for read mapping differed.

## Local eQTL detection and comparison

QTL detection is sometimes referred to as QTL mapping, but we will avoid this terminology to avoid confusion with read mapping. Local eQTLs (often referred to as cis eQTLs) were detected using FastQTL (version 2.184) using 2 Mbps above and below transcription start site

(TSS). 1000 permutations were used to adjust p-values for multiple markers tested and seed 1 was chosen to help reproducibility. Correction for multiple gene testing was performed with R (version 3.4.2) package qvalue (version 2.10.0). The percentage of expressed genes that have a significant local eQTL serves to measures the improvement between the different values used for the mapping parameters tested in the evaluation. The slope (allelic mean difference, representing the direction and strength of allele-specific gene expression) of the linear regression and qvalue of eQTLs from different references were considered similar (unaffected) if they are within less than 5%:

$$\left| \frac{(X_{BXD} - X_{GRCm38})}{average(X_{BXD}, \ X_{GRCm38})} \right| < 0.05,$$

where X refer to slope or qvalue.

### Reference bias

Assuming that D2 alleles are on average as much expressed as B6 alleles, significant eQTLs are as likely to have one or the other allele more expressed. The percentage of skewness of local eQTLs is thus calculated as follows to reflect reference bias (0% indicates no reference bias):

$$\frac{significant \ eQTLs \ with \ negative \ slope - significant \ eQTLs \ with \ positive \ slope}{expressed \ genes} \cdot 100,$$

where significant is defined as FDR < 5%.

### Computational requirements

Some computations were performed on the Wally cluster of the University of Lausanne with the Vital-IT software stack (https://www.vital-it.ch) of the Swiss Institute of Bioinformatics for speed (parallelizing multiple mapping runs) and convenience (having a functional installation of FastQTL software). However, none of the steps require unreasonable memory or computational power, and all softwares used in this study are freely available for reproducibility purpose.

## Results

To improve genetic coherency of RNA-seq read mapping, we explored two alternative strategies to exploit available data in the BXD panel. The first strategy used the two parental strains (B6 & D2) assemblies (Fig 1B "2 parental assemblies"). The second strategy used BXD-specific references obtained from the GRC assembly modified with BXD known and imputed variants (Fig 1B "BXD-specific references"). An intermediate between these two strategies was used for comparison: a D2 reference built from the GRC assembly modified with known D2 variants (Fig 1B "2 parental references"). For each strategy, we evaluated the impact on various downstream steps of the analytical pipeline by quantifying how the strategies affected mappability of the RNA reads (Figs 2A–2C and 3A and 3C), gene expression estimates (Figs 2D and 3B), and eQTLs (Fig 4). In addition, we have evaluated how key mapping parameters influence these results (Figs 2B and 2C and 5).

### Mapping strategy with two parental strains assemblies

To explore the impact of using one reference for all samples despite their genetic differences, we mapped all samples on the classical GRCm38 (B6) genome assembly and on the more recent D2 assembly. We expected that more reads from D2 samples would be uniquely

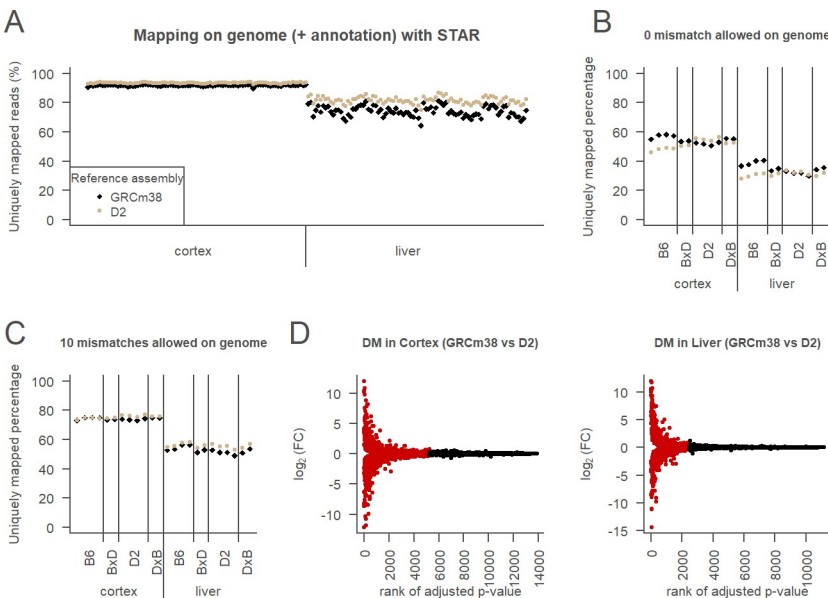

**Fig 2. Two parental assemblies strategy.** A. Mappability of all samples on 2 parental assemblies (samples are mapped on GRCm38: black symbols and on D2 assembly: brown symbols) using permissive mapping setting (STAR default) in cortex (left) and liver (right). Mappability was estimated as the number of uniquely mapped reads expressed as the % of all reads. B. Mappability in samples from the parental strains and their reciprocal F1 offspring (BxD and DxB) on the 2 parental assemblies using restrictive mapping setting allowing 0 mismatches. Same legend than in A. C. Mappability of parental and F1 samples on 2 parental assemblies using restrictive mapping setting but allowing up to 10 mismatches. Same legend than in A. D. Differential mapping (DM) analysis of D2 assembly compared to GRCm38 in the cortex (left) or in the liver (right). Genes are classified as DM genes if FDR adjusted p-value < 0.05 (red) or non DM genes otherwise (black).

mapped on D2 assembly than on the GRCm38 assembly, and that reads from BXD samples would map approximately equally on both parental assemblies. Surprisingly, we observed that the percentage of uniquely mapped reads, used to estimate mappability, was higher for all samples when mapped to the D2 assembly compared to the GRCm38 assembly (Fig 2A), even for B6 samples. We also noticed that mappability differed between the liver and the cortex both in amplitude and in variance. This difference might relate to differences in the preparation of the two tissues for reasons inherent to the tissues, but all the samples passed the quality tests [17]. The liver samples had on average more raw reads than the cortex samples, but the sequencing depth did not seem to explain differences in mappability. It may be that there were more PCR artefacts in the liver reads, so they were either unmapped or multi-mapped which means there were less uniquely mapped reads than in the cortex. Another possibility is that the liver expresses more genes that have regions that are not unique, so more reads are multi-mapped. To further explore the bias for the D2 assembly, we allowed only exact matches. We now observed the expected strain-specificity as B6 samples mapped higher on B6 assembly and D2 samples mapped higher on D2 assembly (Fig 2B). Using up to 10 mismatches (the STAR default for 100 bp reads) but no insertions, deletions and trimming, we lost strain-specificity (Fig 2C). However, the more restrictive mapping setting also importantly reduced the number of uniquely mapped reads (Fig 2A–2C). This raised the question what choice of parameter settings ensures both high read yield and strain specificity (see part "Mapping parameters evaluation" below).

To determine the impact of mapping reference on gene expression, we performed a differential mapping (DM) analysis. The principle is the same as differential expression analysis, but the mapping references are compared instead of different groups or perturbations. Note that

the reads and the values used for the mapping parameters are identical for both references, so differences observed are caused strictly by the reference. More than one third (38%) of genes were affected by the mapping reference (GRCm38 vs. D2) in the cortex and about a quarter (22%) in the liver (Fig 2D). Alignments of the top 4 highly affected genes were visually inspected and revealed variation in the quality of the assemblies and their transcriptome annotation at these precise places. Thus what appeared as differences in gene expression between the two assemblies could in some cases be artefacts and not consequences of genetic variants (examples of artefacts in S2 Fig and S3 Table).

## Customizing reference for D2 and BXD lines

To avoid differences of quality and completeness between B6 and D2 assemblies (S2 Table), we modified the B6 reference assembly using SNPs and indels specific to the D2 strain from dbSNP. We mapped parental and F1 samples with exact matches on both the mm10 assembly and the mm10 assembly modified for D2. The percentage of uniquely mapped reads was increased when the samples were mapped to their corresponding strain reference, compared to the other parental strain reference (Fig 3A). Indeed, D2 samples gained between 4.6 and 5.7% when mapped to the customized D2 reference, whereas B6 samples lost between 3.4 and 5.8%. In contrast, when mapped on the D2 assembly (Fig 2B) D2 samples gained between 0.03% and 3.4% whereas B6 samples lost between 8.5% and 9.8%. The D2 customized reference thus appears more balanced as the gain for D2 samples is closer to the loss for B6 samples. In both cases, the difference between the two parental references was smaller for F1 samples, which is expected for a mix between the two parental strains. To apply the same strategy to

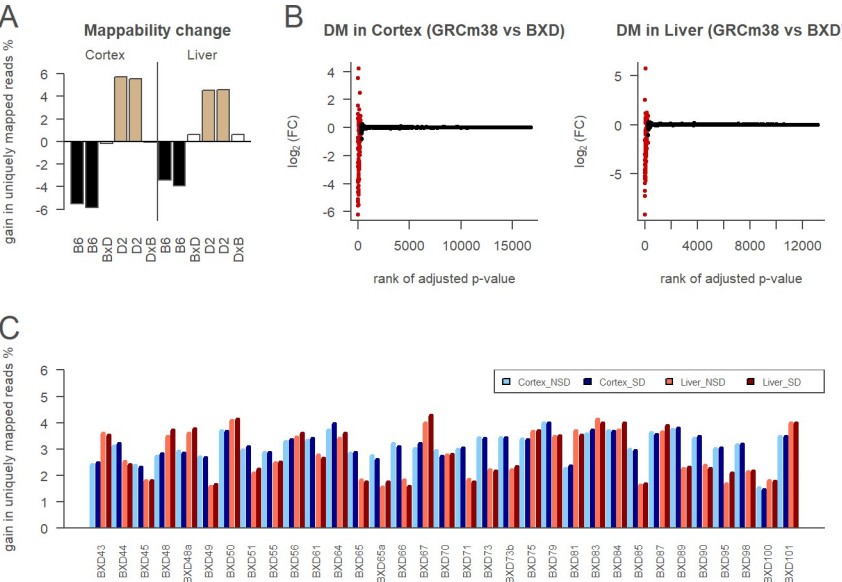

**Fig 3. Line-specific references strategy.** A. Relative mappability of customized D2-specific reference (GRCm38 modified with D2-specific indels and SNVs from dbSNP) compared to GRCm38 on parental and F1 samples with exact matches. Samples are all NSD. Colors indicate genetic of the samples: B6 (black), D2 (light brown), and F1 (white) between B6 and D2 strains. The F1 samples are BxD if the mother is B6 and the father is D2 (as for the BXD lines), or the reverse for DxB. B. Differential mapping (DM) analysis of BXD-specific references compared to GRCm38, in the cortex (left) or in the liver (right). Genes are classified as DM genes if the FDR adjusted p-value < 0.05 (red) or non DM genes otherwise (black). C. Relative mappability of BXD-specific references (GRCm38 modified for each BXD line with GeneNetwork genotypes and imputed variants) compared to GRCm38 on BXD samples with exact matches.

BXD samples, genotypes were imputed using the large amount of D2-specific variants from dbSNP and the BXD specificity of genotypes from GeneNetwork. All BXD samples gained between 1.4 and 4.3% in mappability from having a customized reference by BXD line (Fig 3C). The amplitude of the gain varied among BXD lines and between tissues with cortex samples having globally higher values than liver samples. Note that it was expected the gain to be lower for BXD samples than for D2 samples, since in BXD lines approximately half of the alleles are D2.

To determine the impact of reference customization on gene expression in the BXD lines, we performed a differential mapping analysis. Around 2% of genes are affected by the mapping reference in the cortex and in the liver (Fig 3B).

## Consequences of customization on local eQTL detection

To evaluate the effect of reference customization on estimated biological phenotypes by downstream analysis, we detected local eQTLs using gene expression estimated with the B6 reference or the BXD-specific references. The eQTLs are particularly likely to be influenced since they link gene expression to genetic variants. Significant eQTLs can be seen as a signal-to-noise measure of genetically structured gene expression. The percentage of significant eQTLs was slightly higher (0.1% difference) when using BXD-specific references than when using B6 assembly (Fig 4A). However, this does not necessarily mean that the same eQTLs were detected when using the two different references. The results can differ by the genetic marker associated with the gene expression, the direction of gene expression (whether the gene expression is higher with B6 or D2 allele), or change in the q-value (Figs 4C and S4). When

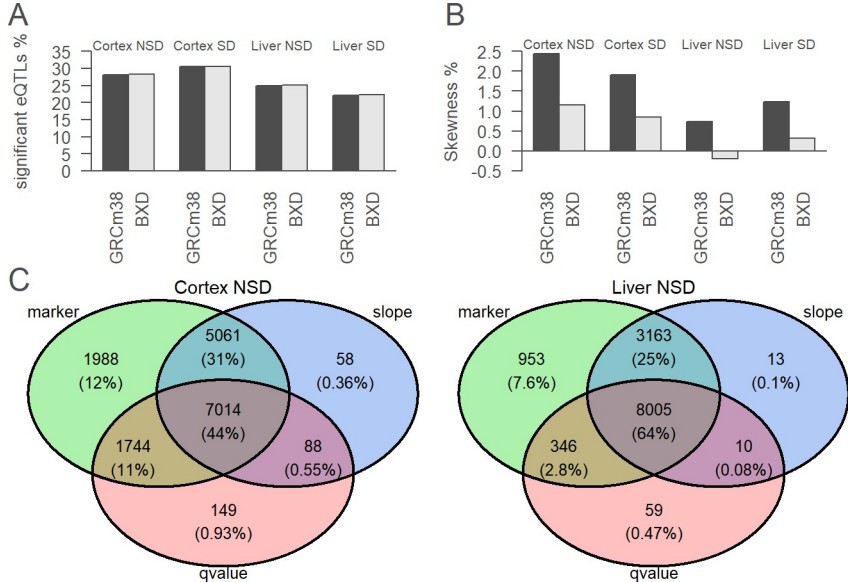

**Fig 4. Consequences of mapping reference at local eQTL level.** A. Percentage of significant (FDR 5%) local eQTLs over all expressed genes with GRCm38 or BXD-specific references. B. Percentage of skewness of significant (FDR 5%) local eQTLs slope over all expressed genes with GRCm38 or BXD-specific references. C. For all expressed genes, the best local genetic marker to explain gene expression was selected. The Venn diagrams represent the overlap of this analysis between GRCm38 and BXD-specific references for the three criteria in cortex NSD (left) or in the liver NSD (right). The marker (in green) indicates changing the reference result in the same genetic marker associated with gene expression. The slope (in blue) is the direction and strength of allele-specific gene expression, it is considered to be overlapping between the references if it varies less than 5%. The qvalue (in pink) is the statistical significance of the marker to gene expression association, it is considered to be overlapping between the references if it varies less than 5%.

considering these 3 variables, mapping reference did not affect 44% of local genetic marker to gene expression association in the cortex and 64% in the liver.

### Reference bias

Next, we wanted to detect a potential reference bias, where reads containing B6 alleles get more easily mapped than those containing D2 alleles or the contrary. In DNA-seq studies, this can be achieved by checking the symmetry of the distribution of allelic ratios at heterozygous loci. In RNA-seq, allele-specific expression can also modify allelic ratio. However, we assumed that globally genes with B6 or D2 alleles are equally expressed. Moreover, since our samples are inbred lines, heterozygous sites are rarer than in other populations, so we compared homozygous alleles of genetically different samples, rather than heterozygous alleles from one sample. The percentage of skewness represents how many local eQTLs deviate from a situation without reference bias (skewness 0%). A positive percentage indicates a B6 bias: more eQTLs with the B6 allele increasing gene expression, whereas a negative percentage indicates a D2 bias: more eQTLs with the D2 allele increasing gene expression. Using the B6 reference shows a reference bias for B6 alleles in all tested tissues and conditions while using BXD-specific references decreased bias (Fig 4B).

### Mapping parameters evaluation

The reference used is not the only thing influencing read mapping. We used the BXD-specific references, and to test which values to use for the more critical mapping parameters of STAR we varied: i) the number of mismatches allowed, ii) the possibility to trim end of reads, iii) the possibility to splice reads, and iv) the use of known transcriptome annotation. The ratio of significant eQTLs to expressed genes (Fig 5) was used for performance optimization as done previously [14,21]. A higher ratio indicates a higher proportion of genetically structured gene expression versus random variations in gene expression. The best settings in both tissues are to use trimming, splicing, and transcriptome annotation and to allow mismatches. The exact maximal number of mismatches differed: 10 mismatches in the cortex (Fig 5A), but only 2 in the liver (Fig 5B). All top settings use existing transcriptome annotation and thus appears to be the more important parameter.

## Discussion

Genomic variations among individuals are the core of genetic studies. Yet it is common practice in the field to use one assembly as reference for all genetically different samples. Here, we improved genetic specificity of read mapping of BXD samples using publicly available data. Our custom BXD-specific references detected proportionally more eQTLs and alleviated reference bias. Below we will discuss the complexity of assessing the analytical design of RNA-seq and the various strategies to integrate genomic variations in transcriptomic analyses.

Although the analysis of RNA-seq data is often regarded as well established, it remains a complex procedure. A great number of factors are involved, going from experimental design (number of replicates, kit for library preparation, sequencing platform, read length) to softwares, functions, and settings used in the different analytical different steps (quality control, mapping, filtering, normalization). We are not being exhaustive about all these aspects, but nevertheless think our observations and considerations on selected features are useful for the community. One of these observations concerned the unanticipated tissue differences in mappability. We were unable to identify supportive literature for this phenomenon and think it merits a more thorough analysis addressing such tissue effects (e.g. using GTEx).

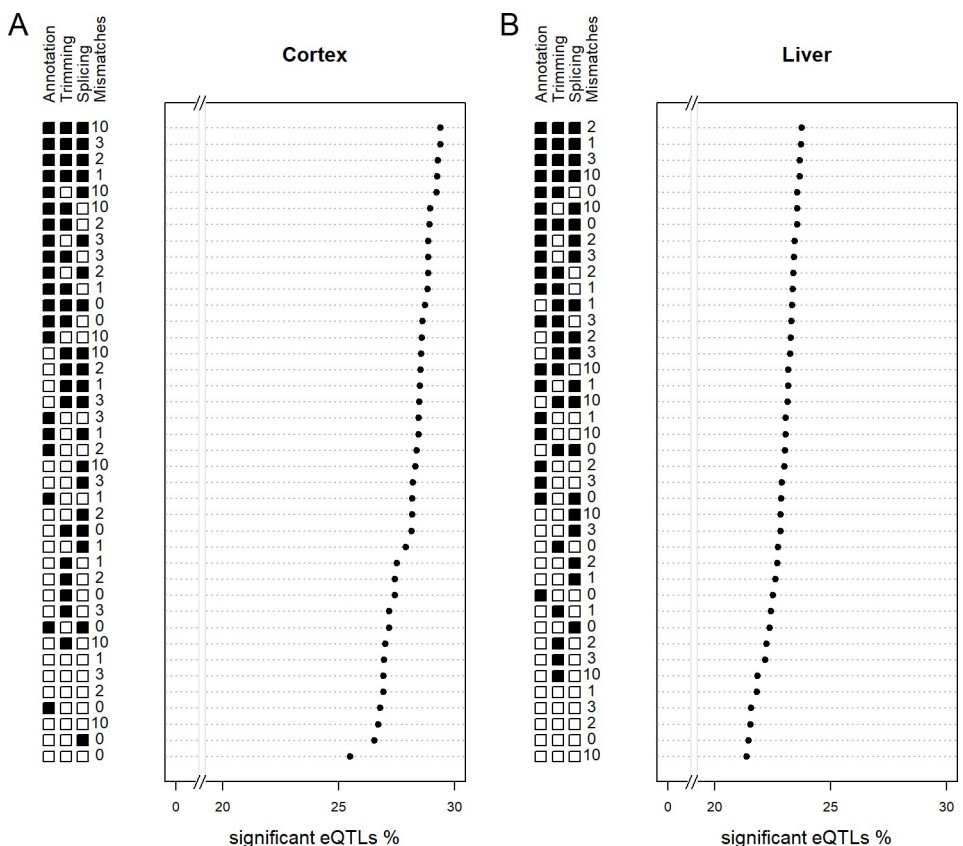

**Fig 5. Evaluating mapping parameters.** A. The performance on local eQTLs of selected mapping settings on cortex samples (average of the NSD and SD conditions) is measured by the percentage of expressed genes that have a significant local eQTL. The BXD-specific references were used. C. As in A but for liver samples.

Another difficulty is that for mapping of real samples, the true location of reads is unknown. The fraction of uniquely mapped reads is used as a mapping statistic because an RNA molecule can only come from one locus. However, this does not guarantee the correctness of mapping of uniquely mapping reads. Importantly, some reads are expected to be correctly classified as multi-mappers because some regions in the genome are identical or very similar (e.g. repeat elements). Moreover, uniqueness can have slightly different meanings depending on the mappers and parameters used, as reads are not necessarily exactly and fully aligned because of mismatches, indels, and sequencing errors especially at the ends of reads (trimming).

Most RNA-seq studies use standard analytical pipelines with default setting, or with slight modifications such as the number of mismatches allowed. Mismatches have the task to compensate for sequencing errors or small unknown variants to give some flexibility in case an exact match is not found. However, the choice of the number of mismatches allowed is rarely given, even though it has been shown in humans that reducing the number of mismatches allowed increased the difference in uniquely mapped reads when using a general versus an individual-specific reference [10]. We showed that the mapping settings have an effect on mappability and also on the local eQTLs. An interesting benchmark was completed on human RNA-seq data in a differential gene expression (DE) pipeline [22]. They compared filtering methods, along with transcriptome reference sets, normalization and DE detection, alignment and counting software. The authors concluded that the optimal filtering threshold depended

on the pipeline parameters. Particularly, the mapping software had the least impact whereas the transcriptome annotation had the higher impact. Also, our study highlighted the importance of transcriptome annotation even if our evaluating measure differed. By focusing on the impact of genetic differences in the reference we could assess different combinations of mapping options. We used the fraction of eQTLs as evaluation measure rather than DE, because we compared B6 versus D2 alleles in a genetic population and not two fixed groups. Indeed, each BXD sample will be considered to be B6 at some loci and D2 at others. A reference bias is likely to influence eQTLs because a variant in a certain gene can modify the mapping of the reads precisely for the samples that have the alternative allele in this region.

An assembly is a global solution because it uses all genomic variants specific to a strain regardless of their size. However, we observed that the technical difference in quality between the two parental assemblies prevents a fair comparison of the genetic difference. In this context it can be noted that we can exclude mix-up of samples as a possible cause of D2 bias in assembly mapping, because it affected all samples: BXD lines, B6 and D2, and because our previous analysis confirmed that this had likely not occurred [18]. Even if transcriptome annotation is likely to be similar in other strains, the different coordinate systems between the assemblies complicate the transfer. Moreover, the actual D2 transcriptome annotation corresponding to the D2 assembly includes manual curation steps that make it hard to update to new releases of the genome assembly or variants. Notably, no study was published using this D2 assembly, except the one from the group that released it [15]. In contrast, our customization of one assembly offers the advantage that the coordinate changes are formalized which allows automatization of transcriptome annotation changes with the same tool used for upgrading versions of an assembly (liftOver). For mapping reads of D2 samples and those of other strains than B6, we currently recommend the use of GRCm38 assembly modified with strain-specific indels and SNVs from dbSNP.

Our custom references combine the specificity of BXD genotypes with the large amount of D2-specific short variants from dbSNP. Importantly, we did not include structural variants (SVs), although many were detected between B6 and D2 strains [23]. SVs can have important phenotypic impacts [24], potentially more than SNVs [25,26]. However, SVs calls will require further efforts in the reporting to ensure the confidence and the format for integration into current workflows. This is due largely to the nature of SVs: their length and large variety implies that the possible number of SVs is greatly superior to that of SNVs, making them less easy to validate and report. Without technologies like long read sequencing and optical genome mapping, those SVs will be very likely inaccessible for mouse models unless an international consortium tackles this issue.

Another limitation is that all murine assemblies are haploid whereas mice are diploid. The diploidy is ignored at the mapping step under the assumption that the genotypes of inbred strains are mostly homozygous. However, the homozygosity and stability of inbred mouse strains is based on a theoretical model that does not consider new mutations [27], although germline mutations are estimated to be between 10 and 30 per generation [28,29]. Unfortunately, the assumption of stability of inbred lines is so strongly anchored in the field that its verification is compromised, because of not searching for heterozygous sites or dismiss them. Indeed, the term heterozygous is sometimes used to call variants uncertain or low quality, and they are always unphased. When mouse assemblies were built, regions with high density of heterozygous sites were used to detect (haploid) assembly errors, ignoring the potential coherence of diploid or polyploid references [15]. A more systematic detection and characterization of heterozygous regions will likely improve the accuracy of transcriptomics studies, particularly for loci with allele-specific expression. However, the read mapping of different possible alleles, which could also be used for not inbred crosses between 2 strains would require a

reconciliation step, as implemented for example for human with paternal and maternal allele [9]. Indeed, every read can come from either one of the two alleles but not from both at the same time. We made an effort to improve the D2 parts of reference to map the BXD samples, however the B6 strain itself is also susceptible to mutations, as confirmed by the occurrence of many B6 substrains, even if it affects only a few genes. Therefore, a complete characterization of genetic variants of the BXD by DNA-sequencing could improve the customization of both D2 and B6 parts of BXD, and therefore enhance resolution of downstream analyses.

## Conclusion

In current genetic studies using the BXD population, genomic variations are paradoxically ignored at the read mapping step, which as we show here causes a reference bias. The genomic variations need to be explicitly integrated in the reference instead of treated as sequencing errors. Our results show the need for a critical evaluation of the RNA-seq pipeline and the development of more complete genomic variants databases to best approximate the genetics of the samples. Most genetic studies with a transcriptomic component in mice and other model organisms can suffer from reference bias, which could be attenuated by assessing and sequencing those strains. The mouse community could follow the drosophila community (http://dgrp2.gnets.ncsu.edu) and sequence genetic reference populations. Our study can serve as a wake-up call for improving the characterization of genomic variations, and as a concrete guide for analyses in BXD and other genetic populations. As RNA-seq analyses are often a starting point to identify one or a few genes that then are studied in more detail in follow-up experiments, it is worth the extra effort to avoid potential bias by not blindly following traditional pipelines.

## Supporting information

**S1 Table. Mouse replicates.**
(CSV)

**S2 Table. Summary statistics of the mouse genome assemblies.** Assembly summary statistics calculated with GAAS toolkit (https://github.com/NBISweden/GAAS).
(CSV)

**S3 Table. Broad classification of DM genes according to the two types of artefacts identified.** S2 Fig provides examples of such artefacts.
(CSV)

**S1 Fig. Consequences of mapping reference transcriptome at read mapping level.** A. Mappability of all samples on 2 parental assembly transcriptomes using STAR permissive mapping setting. B. Pseudo-mappability of all samples on 2 parental assembly transcriptomes using Kallisto. C. Mappability of parental and F1 samples on 2 parental assembly transcriptomes using STAR restrictive mapping setting. D. Mappability of parental and F1 samples on 2 parental assembly transcriptomes using STAR restrictive mapping setting, but up to 10 mismatches.
(TIFF)

**S2 Fig. Examples of artefacts of assemblies sequence and annotation.** A. *Nova2* genomic region in Integrative Genomics Viewer (IGV https://software.broadinstitute.org/software/igv/), as a transcriptome annotation artefact. The coverage is very similar between GRCm38 and D2 assemblies, but the annotation differs, which causes the reads to be counted differently. B. *Gm15564* genomic sequence in IGV (https://software.broadinstitute.org/software/igv/), as an

artefact due to difference in completeness of genome assembly. Many reads map to this region on GRCm38 assembly, but not on D2. It appears that in this region of the D2 assembly there are three unknown nucleotides (with label "N"), which supports the interpretation that it is probably due a difference in assembly quality, and not to a genomic variant.
(TIFF)

**S3 Fig. Genotype imputation workflow.**
(TIFF)

**S4 Fig. Consequences of mapping reference at local eQTL level in SD condition.** A. For all expressed genes, the best local genetic marker to explain gene expression is selected. The Venn diagrams represent the overlap of this analysis between GRCm38 and BXD-specific references for the three criteria in cortex SD. The marker (in green) indicates changing the reference result in the same genetic marker associated with gene expression. The slope (in blue) is the direction and strength of allele-specific gene expression, it is considered to be overlapping between the references if it varies less than 5%. The qvalue (in pink) is the statistical significance of the marker to gene expression association, it is considered to be overlapping between the references if it varies less than 5%. B. Same than A but in the liver SD.
(TIFF)

## Acknowledgments

We thank Mark Ibberson and Brian Stevenson for the helpful discussions and guidance.

## Author Contributions

**Conceptualization:** Nastassia Gobet, Maxime Jan, Paul Franken, Ioannis Xenarios.

**Data curation:** Nastassia Gobet, Maxime Jan.

**Formal analysis:** Nastassia Gobet.

**Funding acquisition:** Paul Franken, Ioannis Xenarios.

**Investigation:** Nastassia Gobet.

**Methodology:** Nastassia Gobet, Maxime Jan, Paul Franken, Ioannis Xenarios.

**Project administration:** Nastassia Gobet, Ioannis Xenarios.

**Resources:** Paul Franken.

**Software:** Nastassia Gobet.

**Supervision:** Maxime Jan, Paul Franken, Ioannis Xenarios.

**Validation:** Nastassia Gobet, Ioannis Xenarios.

**Visualization:** Nastassia Gobet.

**Writing – original draft:** Nastassia Gobet.

**Writing – review & editing:** Nastassia Gobet, Maxime Jan, Paul Franken, Ioannis Xenarios.

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
