## [Decision Letter · Decision Letter 0]

24 Mar 2022

Dear Prof. Xenarios,

Thank you very much for submitting your manuscript "Towards mouse genetic-specific RNA-sequencing read mapping" for consideration at PLOS Computational Biology.

As with all papers reviewed by the journal, your manuscript was reviewed by members of the editorial board and by several independent reviewers. In light of the reviews (below this email), we would like to invite the resubmission of a significantly-revised version that takes into account the reviewers' comments.

We cannot make any decision about publication until we have seen the revised manuscript and your response to the reviewers' comments. Your revised manuscript is also likely to be sent to reviewers for further evaluation.

Sincerely,

Sonika Tyagi

Guest Editor

PLOS Computational Biology

Ilya Ioshikhes

Deputy Editor

PLOS Computational Biology

Reviewer's Responses to Questions

**Comments to the Authors:**

Reviewer #1: In this manuscript Gobet et al examine the impact the choice of reference genome and existing genetic variation has on downstream analyses. It is an interesting study with potential implications for downstream analyses in model organisms. The code is available and the methods are well described. While interesting, I found the manuscript requires clearer descriptions of the experiment, more examples of types of errors, and further discussion about what the results mean for both inbred and outbred organisms.

Major comments:

1) Experimental design: While I eventually was able to understood the design the manuscript would benefit from clearer text and a flowchart describing the analysis. Figure 1 B attempts this but the legend does not sufficiently detail the image. This should be improved and presented at the onset. Further the ‘hybrid’ strategy (BXD-specific references) is poorly described and I am unclear what, if any analyses, were preformed using this strategy.

2) The authors conclude the modified B6 reference is superior relative to the two reference approach however this is potentially just reflecting the lower quality of the D2 reference as you later mention. On L454 you state “However, we showed that the gap in quality and completeness between the two parental assemblies is masking the genetic specificity”. While this may be true, you do not demonstrate this in the paper. In order to make any such conclusions, detailed information on the quality of the two references is required.

3) In Fig S2 you provide a few examples of read mismapping largely due to differences in the assembly and annotation. While useful, it would be better to try to broadly classify the types of issues encountered via a diagram or table. For example, are gene families or repeat elements regions prone to mismapping? Perhaps looking at the cases in Figure 4 that actually change the eQTL slope would be insightful.

4) Many factors are known to have a significant impact on read mappability including the choice of aligner and read length. More discussion is needed on such factors and what (if any) impact they would be expected to have on the results of this study.

5) What is the impact on mappability of SNVs compared to small indels? While SNVs are more numerous indels are more likely to affect mapping. What portion of mismapped reads are attributable to each variant class?

6) You recommend modifying mm10 with strain specific variants to achieve the best results. This is not a trivial undertaking however, does your code offer this functionality?

7) The study describes work on inbred mice however what would your recommendations be for human as an outbred example? Or is your approach only be suitable for inbred organisms?

Minor comments:

1) L129: “Assembly refers to genomic sequences assembled from DNA reads to form chromosomes whereas reference simply refers to the sequences the reads are compared to during read mapping” -> This needs to be clearer.

2) L325 “we modified the B6 reference assembly using SNPs and indels specific to the D2 strain from dbSNP. We mapped parental and F1 samples on these two assemblies with exact matches.” -> what 2 assemblies, mm10 and modified?

3) L332: “D2 samples gained between 0.0% and 3.4%” -> You don’t gain 0% and also why where there no gains in this instance?

4) In Figure 5 you aim to maximise eQTLs. Can you explain why more eQTLs are indicative of better performance?

5) L434 “However, even though it has been shown to importantly impact genetic specificity (Yuan et al., 2015)” -> What exactly is meant by “genetic specificity”? Is it reads uniquely mapped? Regardless, it need to be clearly defined

6) L449: “Indeed, each BXD sample has both B6 and D2 alleles, so the samples compared for each gene are not split the same way, depending on alleles at this locus.” -> What does “depending on alleles at this locus” mean?

I found a few typos / poorly worded sentences throughout. A few are highlighted below although I would suggest a thorough proof read.

Example typos and sentence structures:

L140: “sequences are not containing alternative haplotypes” -> do not contain

L166: “only one BXD lines” -> line

L384: “However, we assumed that globally B6 alleles are equally expressed than D2 alleles. Moreover, since our samples are inbred lines, the heterozygous sites are greatly rarer” -> Two errors here

L427: “However, this does not guaranty” -> guarantee

L428: “due to e.g. redundancy in the genome” -> not sure what this means

Reviewer #2: This manuscript describes the importance of the reference genome chosen when aligning short read data for subsequent analysis in RNAseq experiments. The insight from the recombinant inbred strains is particularly interesting. The authors present information of how this may skew findings, especially the results of the differential mapping analysis (Fig 2D and 3B). However, some basic analyses need to be performed to increase the credibility of the results. Especially surrounding the mappability analyses. Also, the manuscript has significant problems in execution, namely the methods. Some of the points raised in the discussion could actually be quickly tested by the authors with existing data.

Methods:

The methods are not publication-quality and are confusing. These need to be re-written for improved understandability and credibility.

Results:

At line 287, the authors note surprising deviations from their expectations of mappability of reads from B6, D2 and BXD lines. One possible interpretation that is not addressed is the possible mixups with individual mice and/or tubes when the original tissue samples were collected. Also, we have seen cross-contamination of samples introduced at sequencing facilities, even including the identification of reads from different mammalian species. The transcriptomic data is from pooled samples from many individual mice and an accidental inclusion of one or mice from the wrong strain into a pooled sample would go some way to explain the results the authors obtained. These sample mix-ups a very common, even in the largest and most automated genome centres. It would be possible to search for reads that contain genotypes that are specific to either B6 or D2 and count the relative presence of these in each sample pool. The presence or absence of contaminating reads (potentially introduced even from the sequencing step) would be instructive to the following results.

On line 290 the authors also describe mappability differences between the cortex and liver samples in the same lines. That mappability should vary between tissues from the same organisms points to a technical difference in the preparation of the samples. The reasons for the differences may be trivial, but the authors need to address this further. For example, are differences in sequencing depth or increased sequencing error important for the mappability differences that are a main focus of the manuscript?

The mappability of reads seems to have been assessed in two ways, i) exact matches, and ii) up to ten mismatches. Both might be problematic: Generally, the rate of short-read sequencing error is ~1 nucleotide per 100, in the 100bp reads they have obtained, most of them will have a sequencing error. So, requiring exact matches will be too strict for many otherwise-mappable reads. However, allowing 10 mismatches will not resolve reads that could be mapped to multiple members of gene families. Given the importance of this to the following analysis, some optimisation of the allowed mismatches in alignment is essential. As would be inclusion of comparison with a different aligner. The “Mapping parameters evaluation” section is interesting, but does not address the need to optimise the number of allowed mismatches.

Discussion:

On line 438 the authors note the need for more investigation of tissue effects and point to existing data from GTex. Given the relevance of this to the current manuscript, potentially the authors might judiciously choose a small number of example datasets from GTex that include cortex/liver comparisons – and investigate these for tissue effects themselves and include this in the manuscript.

On line 479 of the discussion, the point is made about the genetic drift of inbred lines (10 to 30 germline mutations per generation) and the relevance this has to mappability. Which is true, yet de novo variation accumulated since the production of a reference sequence seems a red herring. The variation mentioned is unlikely to be problematic for read-mapping so soon after the reference genomes have been sequenced. Even after a decade and assuming a very high 10 generations per year, this is only 10x10x30 = 3000 de novo variants distributed among 3 gigabases of genome. This may very likely cause phenotypic divergence. Yet, for alignment of RNAseq data, this will be only around 30 coding de novo variants, hence only a little more than one 1 in 1000 genes are likely to harbour a variant that has occurred since the production of the reference sequence for the same strain.

**Have the authors made all data and (if applicable) computational code underlying the findings in their manuscript fully available?**

Reviewer #1: None

Reviewer #2: Yes

PLOS authors have the option to publish the peer review history of their article (what does this mean?). If published, this will include your full peer review and any attached files.

Reviewer #1: No

Reviewer #2: No
---

## [Decision Letter · Decision Letter 1]

18 Jul 2022

Dear Prof. Xenarios,

Thank you very much for submitting your manuscript "Towards mouse genetic-specific RNA-sequencing read mapping" for consideration at PLOS Computational Biology. As with all papers reviewed by the journal, your manuscript was reviewed by members of the editorial board and by several independent reviewers. The reviewers appreciated the attention to an important topic. Based on the reviews, we are likely to accept this manuscript for publication, providing that you modify the manuscript according to the review recommendations.

Sincerely,

Sonika Tyagi

Guest Editor

PLOS Computational Biology

Ilya Ioshikhes

Deputy Editor

PLOS Computational Biology

[LINK]

Reviewer's Responses to Questions

**Comments to the Authors:**

Reviewer #1: My concerns have been adequately addressed

Reviewer #2: Uploaded as an attachment

**Have the authors made all data and (if applicable) computational code underlying the findings in their manuscript fully available?**

Reviewer #1: Yes

Reviewer #2: Yes

PLOS authors have the option to publish the peer review history of their article (what does this mean?). If published, this will include your full peer review and any attached files.

Reviewer #1: No

Reviewer #2: No

Figure Files:

Data Requirements:

Reproducibility:

References:

---

## [Editor Report · Decision Letter 2]

7 Sep 2022

Dear Prof. Xenarios,

We are pleased to inform you that your manuscript 'Towards mouse genetic-specific RNA-sequencing read mapping' has been provisionally accepted for publication in PLOS Computational Biology.

Best regards,

Sonika Tyagi

Guest Editor

PLOS Computational Biology

Ilya Ioshikhes

Section Editor

PLOS Computational Biology

---

## [Editor Report · Acceptance letter]

22 Sep 2022

PCOMPBIOL-D-22-00085R2 

Towards mouse genetic-specific RNA-sequencing read mapping

Dear Dr Xenarios,

I am pleased to inform you that your manuscript has been formally accepted for publication in PLOS Computational Biology. Your manuscript is now with our production department and you will be notified of the publication date in due course.

With kind regards,

Zsanett Szabo
